# CREATIVE STYLE TRANSFER

## ABSTRACT

Generating novel artistic styles from a single style is a formidable challenge for for traditional style transfer techniques, which typically focus on emulating the provided style without introducing fresh and meaningful elements. In this paper, we propose a creativity process for producing new and meaningful artistic styles, called creative style transfer (CSFer). We first introduce a neural permutation network (PerNet) to rearrange the feature maps of a single style image, thereby adapting them to the feature maps of a content image, resulting in the desired stylization content. Essentially, this permutation process enriches the bases of the single style within a high-dimensional feature space, departing from the conventional linear combination of multiple styles. To gauge the quality of our stylized content, we leverage metrics encompassing content structure, style perception, and artistic aesthetics. These metrics enable us to assess our stylized content in comparison to the output produced by traditional style transfer methods. In the training phase, PerNet learns to generate high-quality stylized content by randomly sampling permutation matrices that yield high-quality stylization outcomes. Experimental results demonstrate that our CSFer can create novel and original stylization outcomes. Furthermore, CSFer exhibits robust generalization capabilities by simply inserting the PerNet into the style transfer methods.

## 1 INTRODUCTION

Developing computer vision techniques to create human-level visual art is a long-term goal within the rich history of artificial intelligence and art. One fundamental feature of this endeavor is creativity, which is defined as the process of producing novel and meaningful artifacts while evaluating them simultaneously (Boden, 1998; Cetinic & She, 2022; Boden, 2004). Recently, a popular neural style transfer (NST) technique has been introduced to achieve this goal by transferring the artistic style of one image to another in many successful applications, such as creating artistic images (Gatys et al., 2016), videos (Chen et al., 2017), and 3D scenes (Huang et al., 2021). Although NST has produced many impressive visual artworks (Huang et al., 2017; Chen et al., 2020; Mu et al., 2022), it still struggles to generate truly creative outputs since these methods only imitate the style image, lacking novelty and meaning. Therefore, we are searching for a method that can produce creative artistic styles using a single style image and transfer it to a content image.

Most style transfer methods aim to create new artistic styles but often fall short in terms of novelty and surprise. For instance, style interpolation techniques (Huang & Belongie, 2017; Park & Lee, 2019; Ali et al., 2023) rely on linearly combining the features of several style images with normalized weights. More recent approaches such as style-former (Wu et al., 2021) and ArtINS (Xie et al., 2022) have attempted to increase the diversity of style bases by incorporating FastICA (Hyvarinen, 1999) algorithms, but they still rely on linear combinations of used styles to generate their mixed styles. Unfortunately, despite these efforts, the stylized images produced by these methods are often predictable and fail to deliver novel and surprising results since all the styles are already known and the linear combinations cannot create new style bases from the used styles.

Furthermore, in theory, when a substantial number of style bases are employed, the interpolation technique can generate a wide range of styles through linear combinations. Nevertheless, this process can become computationally demanding when searching for the optimal coefficients. For example, ArtINS (Xie et al., 2022) typically utilizes approximately ten distinct style components in practice.

To tackle this issue, we adopt a streamlined approach by utilizing a deep neural network to compute a style's feature vector, which in turn serves as the foundation for generating a fresh style feature vector. This novel approach draws inspiration from permutation theory, leveraging it to rearrange the style features effectively. By applying permutation to the index set of a feature vector, we can generate diverse feature vectors, thereby creating new stylistic variations. For instance, a vector like $[0, 0, 0, 1, 1, 1]$ can be transformed into a new vector $[0, 1, 0, 1, 0, 1]$ through the use of a permutation matrix. For a feature space comprising of $m$ dimensions, there exists a wealth of potential permutations, specifically $\frac{m(m-1)}{2} - 1$, enabling the creation of numerous novel stylistic outcomes. Following this, we leverage a style perception distance technique (Li et al., 2017a), and an aesthetic assessment method (Yi et al., 2023) to select the most innovative and meaningful style outcomes. This combined approach surpasses the limitations associated with traditional NST methods, enabling the discovery of creative styles from the provided style reference.

In this work, we develop a learnable creative process for the style transfer, named **C**reative **S**tyle Trans**Fer** (**CSFer**), which is shown in Figure 1. Given an artistic style image and a content image, we formulate this process to alternate between producing meaningful styles, which are surprising to the given style, and evaluating their novelties of the stylized content images. Firstly, we learn a neural permutation network (PerNet) to predict the permutation matrix for sorting means and variances of feature maps of the style image, and then use them to adjust those of the content image for producing the newly stylized content. Secondly, we assess the uniqueness of the newly stylized content using a combination of content structure, style perception, and aesthetic metrics, *e.g.*, structural similarity (SSIM) (Wang et al., 2004), style perception distance (SPD) (Li et al., 2017a), and style-specific art assessment network (SAAN) score (Yi et al., 2023). These metrics enable us to highlight distinctions in semantic meaning and structural information between our newly stylized content and the traditional stylized content produced by conventional NST methods like AdaIN (Huang & Belongie, 2017). Thirdly, we train the PerNet by randomly sampling a permutation matrix to satisfy the newly and meaningfully stylized content image. Overall, our contributions are summarized as follows:

- We propose a neural permutation method that sorts the means and variances of feature maps from a single style image, enabling us to capture the distinctiveness and surprise elements inherent to the style image. This approach extends the imitation boundaries of traditional NST methods by generating fresh stylistic feature bases from the single style image.

- By combining the permutation method, we explore a creative style transfer approach to produce novel and meaningful artworks. To the best of our knowledge, we are the first to efficiently and skillfully fuse creativity and style transfer.

- Experiments demonstrate that our method outperforms the creativity or diversity of the stylized styles generated by the existing style transfer techniques, and also exhibits robust generalization capabilities with a straightforward integration process. (see Figures 3 and 6) .

## 2 RELATED WORK

**Artistic style transfer** starts from the pioneered ideology (Gatys et al., 2015) and has become an attractive research topic for both academy and industry since its variants (Li et al., 2019; xin Cheng et al., 2021; Jing et al., 2020) are introduced from single style to multiple/arbitrary style, and produce favorable visual results in many applications, *e.g.*, artistic image/video/text (xia Zou et al., 2021; Chen et al., 2020; Deng et al., 2021; Yang et al., 2019; Zhang et al., 2019; Hong et al., 2021; Kwon & Ye, 2022; Kotovenko et al., 2021; Yang et al., 2022b; Wen et al., 2023; Xu et al., 2023), animation (Siarohin et al., 2019; Aberman et al., 2020; Siarohin et al., 2021; Tianxin et al., 2022), photorealism (Yoo et al., 2019; Xia et al., 2020), shape deformation (Kim et al., 2020; Liu et al., 2021b), and industrial design (Jiang et al., 2022; Yang et al., 2022a).

Recently, a number of works try to increase the diversity of the style using different methods. For example, Style-mixer (Huang et al., 2019; Liu et al., 2021a) introduces a multi-level feature concatenation and a patch attention to achieve better semantic correspondences and preserve richer style details. A diversity loss (Li et al., 2017b) is presented to allow the feed-forward networks to generate diverse outputs. Deep feature perturbation (DFP) (Wang et al., 2020) uses an orthogonal random noise matrix to perturb the deep image feature maps. Moreover, diverse image style transfer (Chen et al., 2021b) enforces an invertible cross-space mapping to achieve significant diversity. Spatial

control (Huang & Belongie, 2017; Gatys et al., 2017; Chiu & Gurari, 2020) is to transfer different styles to different regions of the content image using masks and segmentation. In addition, mentioned in the introduction, style interpolation (Park & Lee, 2019; Liu et al., 2021a) is to interpolate a set of style features with corresponding weights. Style-former (Wu et al., 2021) are based on transformer to find global composition of a finite set of the style codes. Style-discovery (Xie et al., 2022) also builds a set of style components to linearly combine them for discovering new artistic styles.

Our proposed approach differs significantly from previous works that use a linear combination of style components. Instead, we introduce a novel neural permutation method that operates on the features of a single style image, producing meaningful and surprising stylization results that stand out from previous works which often fall within similar stylistic boundaries.

**Creating arts** still has two other important technologies in addition to the artistic style transfer. The first is creative adversarial networks (CAN) (Elgammal et al., 2017), which extends the capabilities of generative adversarial networks (GAN) (Goodfellow et al., 2014) to generate art that deviates from established styles while still adhering to the distribution of art. The second is the creative decoder (Das et al., 2020), which enhances the decoder of variational autoencoder (VAE) (Kingma & Welling, 2014) to generate creative art by using a sampling, clustering, and selection strategy to capture neuronal activation patterns. Moreover, (Cintas et al., 2022) applies group-based subset scanning on node activations from internal layers by integrating GAN and creative decoder. Unlike the two techniques, our approach involves the use of a permutation matrix to sort the means and variances of feature maps in the style image as they rely on style classification and ambiguity losses, and neuronal activation patterns, respectively. Furthermore, our approach outperforms both CAN and creative decoder in producing exceptional creative artworks.

## 3 METHODOLOGY

In this section, we propose a creative style transfer (CSFer) for generating distinctive and meaningful stylized results, as depicted in in Figure 1. Specifically, we present a neural permutation module with a deep network to create new feature maps of the style image. Subsequently, we design a novelty evaluation scheme to assess the novelty of the stylistic outcomes for sampling a meaningful style. Finally, PerNet is trained using randomly sampling permutation matrices with the established evaluation criteria. Before delving into our method, we first present a new style transfer problem.

### 3.1 A NEW STYLE TRANSFER PROBLEM

Standard NST method produces a stylized content image by transferring the style of the reference image into the content image. The resulting stylized image is predominantly influenced by the chosen style image. However, traditional NST techniques lack the capacity for creative artistic expression, as they primarily imitate the style image. This gives rise to a significant challenge, which we refer to as *Creative Style Transfer*, defined as follows:

> *The goal of creative style transfer is to imaginatively infuse a fresh and meaningful style from the style image into the content image.*

Mathematically, given a style image $I_s \in \mathbb{R}^{n \times m \times 3}$ and a content image $I_c \in \mathbb{R}^{n \times m \times 3}$, they are transformed into distinct feature maps $f_s \in \mathbb{R}^{H \times W \times K}$ and $f_c \in \mathbb{R}^{H \times W \times K}$ via an encoder network $E$, such as the pre-trained VGG net (Karen Simonyan, 2015), denoted as $f_s = E(I_s)$ and $f_c = E(I_c)$. Subsequently, a new content feature $\overline{f}_c$ is derived by transferring the style information of $f_s$ into $f_c$, denoted as $\overline{f}_c = \text{Transfer}(f_s, f_c)$, using techniques like AdaIN (Huang & Belongie, 2017), LST (Li et al., 2019) and ArtINS (Xie et al., 2022). Finally, a decoder $D$ is employed to map the transformed feature back to the image space, generating the stylized outcome $O$, denoted as $O = D\left(\overline{f}_c\right)$. In this transformation process, *how to create a fresh and meaningful outcome O?*

This problem can be further subdivided into two key aspects: creating a new and meaningful style from the single style image, and transferring it into the content image. While existing style transfer methods can be employed for the style transfer step, our primary focus lies in the inventive generation of new and meaningful styles from the single style image.

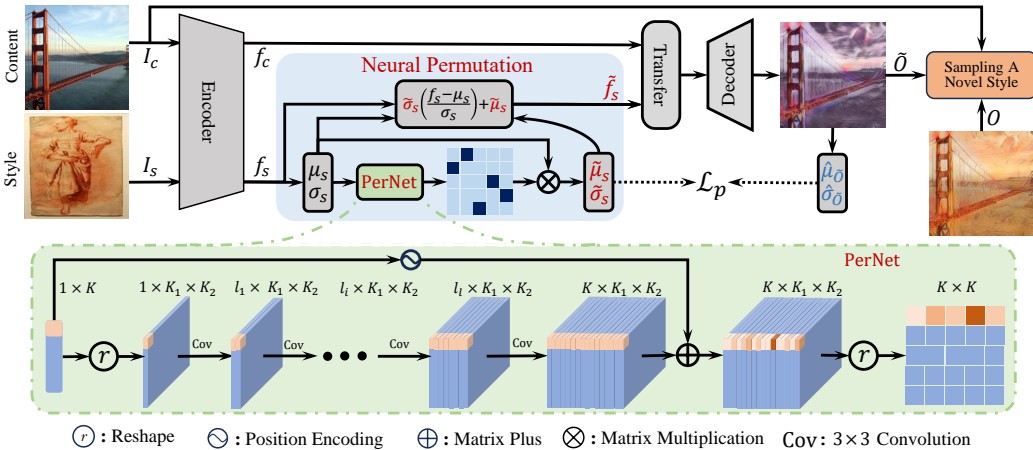

Figure 1: The framework of our creative style transfer. The upper picture illustrates the style transfer process alongside a sampling module that employs content structure, style perception, and aesthetic evaluations The down picture showcases the architecture of the neural permutation network (PerNet).

To tackle this issue, we present a neural permutation mechanism to produce new style feature from the single style image for generating fresh styles. Additionally, we introduce a novelty evaluation criterion to sample distinctive styles. Lastly, we utilize two artistic image aesthetics evaluation techniques, namely SAAN (He et al., 2022) and TANet (Yi et al., 2023), to gauge the quality of the newly generated styles, thereby demonstrating the significance of our style creation approach.

## 3.2 NEURAL PERMUTATION MECHANISM

Here, we introduce a neural permutation mechanism designed to rearrange the means and variances of style feature maps for the creation of new styles, as illustrated in Figure 1. We begin with the style image $I_s \in \mathbb{R}^{n \times m \times 3}$, which is transformed into feature maps $f_s$ through an encoder network $E$, $f_s = E(I_s)$. The means and variances of the style image features $f_s$ are defined as follows:

$$\mu_s = \mu(f_s), \sigma_s = \sigma(f_s), \quad \mu_s, \sigma_s \in \mathbb{R}^K, \tag{1}$$

where $\mu(\cdot)$ and $\sigma(\cdot)$ indicate the computation of the mean and standard deviation of the image feature map across different channels, with $K$ representing the number of channels.

**Neural Permutation.** Our objective is to sort the feature mean and variance, $(\mu_s, \delta_s)$, from the style image $I_s$ to capture different mean and variance characteristics for creating new style features. To achieve this, we design a neural permutation network (PerNet) that operates on the means $\mu_s$ to predict a permutation matrix $P \in \mathbb{R}^{K \times K}$. The sorted mean and variance can then be obtained through matrix-vector multiplication as follows:

$$\widetilde{\mu}_s = P\mu_s, \quad \widetilde{\sigma}_s = P\sigma_s \quad \text{with} \quad P = \text{PerNet}(\mu_s; \phi). \tag{2}$$

where $\phi$ represents the network parameters of the designed PerNet, and the crimson letters correspond to Figure 1. Subsequently, we align the permuted channel-wise mean and variance to match those of the style feature $f_s$, resulting in the new style feature $\widetilde{f}_s$ obtained through an affine transformation:

$$\widetilde{f}_s = \widetilde{\sigma}_s \left( \frac{f_s - \mu_s}{\sigma_s} \right) + \widetilde{\mu}_s. \tag{3}$$

In this transformation, the new feature scales the normalized content input with $\widetilde{\sigma}_s$, and shifts it with $\widetilde{\mu}_s$. Finally, following the transfer function and decoder in the standard style transfer process, we can generate a new stylized outcome $\widetilde{O} = D(\widetilde{f}_c)$, where $\widetilde{f}_c = \text{Transfer}(\widetilde{f}_s, f_c)$.

**Architectures of PerNet.** Due to difficulty of learning the binary matrix, we utilize the Gumbel-Sinkhorn Softmax method (Adams & Zemel, 2011; Mena et al., 2018) for the purpose of learning the permutation matrix. As shown in Figure 1, the architecture of PerNet is defined as follows:

$$\text{PerNet}(\mu_s; \phi) = N_c \left( N_r \left( \frac{\exp(\text{Net}(\mu_s; \phi))}{\tau} \right) \right), \tag{4}$$

where $\text{Net}(\mu_s; \phi) \in \mathbb{R}^{K \times K}$ represents a feed-forward convolutional network with network parameters $\phi$ and mean inputs $\mu_s$, the symbol $\tau$ denotes the temperature parameter, $N_c(M)$ and $N_r(M)$ denote the column and row-wise normalization operators applied to a matrix, respectively.

The architecture of $\text{Net}(\mu_s; \phi)$ comprises two branches. The first branch involves reshaping the mean inputs $\mu_s \in \mathbb{R}^{1 \times K}$ into a $1 \times K_1 \times K_2$ mean map, where $K = K_1 \times K_2$. Subsequently, convolutional operations are applied to obtain feature maps from $1 \times$ to $l_i \times$ and $K \times K_1 \times K_2$ by increasing the number of channels step by step, where $l_i$ is the number of channels at $i$th layer. Recognizing the significance of position information alongside the means, the second branch simultaneously incorporates a position encoding into the features, defined by (Vaswani et al., 2017):

$$\text{PE}(pos, 2i) = \sin\left(\frac{pos}{10000^{2i/d_{\text{model}}}}\right), \quad \text{and} \quad \text{PE}(pos, 2i+1) = \cos\left(\frac{pos}{10000^{2i/d_{\text{model}}}}\right), \quad (5)$$

where $pos$ denotes the position of a value within the entire vector, where $0 \leq pos < K$, and $d_{\text{model}}$ represents the positional arrangement information for the row, with the variable $i$ ranging from 0 to $d_{\text{model}} - 1$. To encode these positions, we utilize the Eq. 5: left equation for even values of $i$ and right equation for odd values of $i$. Next, we combine the results from these two branches and incorporate the resulting position embeddings, denoted as PE, into the feature vector $f_P$. This vector is then reshaped into a permutation matrix of dimensions $K \times K$, yielding an output matrix $P$. During the testing phase, it's important to note that we convert the soft permutation into a hard permutation matrix using the Hungarian algorithm as described in (Kuhn, 1955).

### 3.3 Sampling a Novel Style from Newly Stylized Outcomes

Building upon the permutation mechanism discussed earlier, this method can produce a set of fresh stylized images, denoted as $\widetilde{\mathcal{O}}$. This is achieved by utilizing a collection $\mathcal{P}$ consisting of $M$ randomly generated permutation matrices and the affine transformation presented in Eq. 3 for both the style image $I_s$ and the content image $I_c$. In this section, we introduce a novel sampling approach to choose the most suitable style from the set $\widetilde{\mathcal{O}}$. The primary concept involves comparing the newly generated stylized result $\widetilde{O}$ with the standard stylized result $O$ as well as the content image $I_c$. This selection process encompasses a balanced sampling technique along with an aesthetic evaluation.

**Sampling a subset $\widetilde{\mathcal{O}}_{\text{sub}}$ from $\widetilde{\mathcal{O}}$ utilizing SSIM and SPD scores.** (please refer to Appendix A.) Our aim is to sample a stylized output $\widetilde{O}$ that preserves the structural characteristics of $I_c$ while exhibiting a more diverse style. To ensure the preservation of structural features, we employ the Structural Similarity Index (SSIM) (Wang et al., 2004) to calculate the SSIM score between $\widetilde{O}$ and $I_c$, ensuring they are suitably similar. To promote a wider style diversity, we employ the Style Perception Distance (SPD) (Li et al., 2017a) to compute SPD scores between $\widetilde{O}$ and $I_c$ as well as between $\widetilde{O}$ and $O$. We then select outcomes with higher SPD scores. This sampling process is defined as follows:

$$\widetilde{\mathcal{O}}_{\text{sub}} = \left\{ \widetilde{O} \mid \alpha \leq \text{SSIM}(\widetilde{O}, I_c) \ \& \ \beta_1 \leq \text{SPD}(\widetilde{O}, I_c) \ \& \ \beta_2 \leq \text{SPD}(\widetilde{O}, O), \ \widetilde{O} \in \widetilde{\mathcal{O}} \right\}, \quad (6)$$

where $\alpha$, $\beta_1$ and $\beta_2$ represent hyper-parameters that control the boundaries of these conditions.

**Selecting the optimal $\widetilde{O}_{\text{opt}}^{(I_s, I_c)}$ from $\widetilde{\mathcal{O}}_{\text{sub}}$ using aesthetic assessment.** To determine the most aesthetically pleasing style, we leverage the Style-Specific Art Assessment Network (SAAN) score (Yi et al., 2023) for aesthetic quality assessment. We select the optimal aesthetic style by maximizing the following objective:

$$\widetilde{O}_{\text{opt}}^{(I_s, I_c)} = \arg\max_{\widetilde{O} \in \widetilde{\mathcal{O}}_{\text{sub}}} \left\{ \text{SAAN}(\widetilde{O}) \right\}, \quad (7)$$

where $\text{SAAN}(\widetilde{O})$ directly predicts the aesthetic score of the image $\widetilde{O}$. SAAN is trained using a dataset of artistic images from Boldbrush, with human-labeled scores collected from online users.

In Figure 2, we illustrate a practical sampling process employing the AdaIN technique (Huang & Belongie, 2017). This process involves generating a set denoted as $\mathcal{P}$ consisting of $1,000$ random

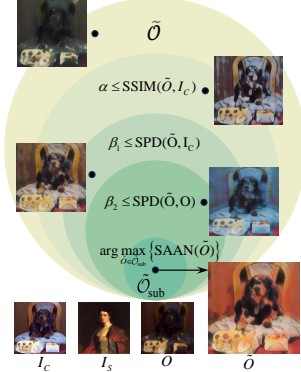

Figure 2: Sampling process for creating stylized images.

Table 1: Aesthetic evaluation of style transfer methods on three datasets: WikiArt, CAN and CD. Note that ArtINS and ArtINS+CSFer only use the images in Figure 3 to compute the scores.

| Methods | Aesthetic SAAN Scores (Yi et al., 2023) | | | Aesthetic TANet Scores (He et al., 2022) | | |
|---|---|---|---|---|---|---|
| | WikiArt | CAN | CD | WikiArt | CAN | CD |
| Style Images | 4.519±0.205 | 4.585±0.047 | 4.723±0.312 | 5.119±0.078 | 5.076±0.040 | 4.156±0.007 |
| DFP | 4.674±0.184 | 4.575±0.065 | 4.695±0.093 | 5.046±0.109 | 4.604±0.038 | 4.530±0.108 |
| AdaAttN | 3.884±0.093 | 4.517±0.049 | 4.659±0.019 | 5.253±0.051 | 4.784±0.042 | 4.810±0.051 |
| ArtFlow | 4.485±0.089 | 4.483±0.044 | 4.333±0.056 | 4.700±0.039 | 4.898±0.032 | 4.378±0.022 |
| StyTr$^2$ | 4.631±0.103 | 4.459±0.037 | 4.598±0.068 | 5.282±0.057 | 4.989±0.045 | 5.171±0.032 |
| AdaIN | 4.634±0.112 | 4.556±0.066 | 4.441±0.046 | 5.360±0.036 | 4.453±0.021 | 4.296±0.062 |
| **Our CSFer** | 4.675±0.069 | 4.683±0.069 | 4.458±0.154 | **5.444±0.027** | 4.889±0.042 | 4.748±0.058 |
| MST | 3.885±0.093 | 4.700±0.073 | 4.740±0.071 | 4.331±0.119 | 5.517±0.028 | 5.373±0.073 |
| **MST+CSFer** | 3.888±0.095 | **4.746±0.063** | **4.776±0.097** | 4.338±0.123 | **5.521±0.026** | **5.458±0.028** |
| LST | 4.621±0.081 | 4.612±0.040 | 4.485±0.181 | 4.810±0.044 | 4.845±0.012 | 4.539±0.041 |
| **LST+CSFer** | 4.704±0.044 | 4.722±0.024 | 4.686±0.032 | 4.839±0.029 | 4.863±0.021 | 4.567±0.007 |
| IEC | 4.532±0.077 | 4.563±0.024 | 4.434±0.097 | 4.952±0.052 | 5.103±0.032 | 4.442±0.016 |
| **IEC+CSFer** | 4.599±0.033 | 4.534±0.030 | 4.537±0.023 | 4.862±0.039 | 4.875±0.024 | 4.762±0.036 |
| ArtINS | 5.053±0.341 | - | - | 4.118±0.051 | - | - |
| **ArtINS+CSFer** | **5.112±0.376** | - | - | 4.144±0.027 | - | - |

permutation matrices, subsequently leading to the creation of a stylized image set named $\widetilde{\mathcal{O}}$. Using the Eq. 6, we then extract a subset denoted as $\widetilde{\mathcal{O}}$sub from $\widetilde{\mathcal{O}}$ with the dual objective of preserving clear content information while introducing distinct style characteristics. Finally, we identify the optimal result $\widetilde{O}_{\text{opt}}^{(I_s, I_c)}$ by employing the Eq. 7.

### 3.4 TRAINING LOSS

Using the sampling method described above, we gather the optimal outcomes $\widetilde{O}_{\text{opt}}^{(I_s, I_c)}$ from a variety of style and content image pairs $(I_s, I_c)$ by employing the AdaIN technique (Huang & Belongie, 2017). Subsequently, we can generate numerous pairs of original style images and creatively modified style images as $(I_s, \widetilde{O}_{\text{opt}}^{(I_s, I_c)})$. We then calculate their respective feature means, denoted as $(\mu_s, \widehat{\mu}_{\widetilde{O}})$, where $\mu_s = \mu(E(I_s))$ and $\widehat{\mu}_{\widetilde{O}} = \mu(E(\widetilde{O}_{\text{opt}}^{(I_s, I_c)}))$ using the pre-trained VGG net (Karen Simonyan, 2015). These data points collectively form a dataset denoted as $\mathcal{D}$. Using this dataset, we can train the PerNet model in Eq. 2 within a supervised setting. The training loss is defined as follows:

$$\mathcal{L}_P = \frac{1}{|\mathcal{D}|} \sum_{(\mu_s, \widehat{\mu}_{\widetilde{O}}) \in \mathcal{D}} \left\| \text{PerNet}(\mu_s, \phi)\mu_s - \widehat{\mu}_{\widetilde{O}} \right\|_2^2. \tag{8}$$

## 4 EXPERIMENT

### 4.1 EXPERIMENTAL SETTINGS

**Training and Datasets.** We train our model (*i.e.*, the decoder network and the PerNet network) though two stages using Wikiart (Phillips & Mackintosh, 2011) as style images and MS-COCO (Lin et al., 2014) as content images. In the first stage, following the setting (Huang & Belongie, 2017), we build about 80, 000 training style and content image pairs to train our the decoder network. The training setting is similar to the AdaIN (Huang & Belongie, 2017) by using the adam optimizer (Kingma & Ba, 2015) and a batch size of 16 content and style image pairs, resize them to the smallest size $512 \times 512$, and randomly crop regions of size $256 \times 256$.

In the second stage, we randomly choose 4400 content and style image pairs from MS-COCO (Phillips & Mackintosh, 2011) and Wikiart (Lin et al., 2014). For each image pair, we use the novelty sampling scheme in Eqs. 6 and 7 to sample 1000 meaningful stylized images, and select one of them as an optimally stylized image. Then, these 4400 image pairs lead to 4400 feature mean pairs using the pre-trained VGG net (Karen Simonyan, 2015). We still use the Adam optimizer (Kingma & Ba, 2015) to train the PerNet network with a gradual decline in learning rate and a batch size of 25 feature mean pairs. We can test arbitrary other style images. When $K = 512, K_1 = 16, K_2 = 32$, the PerNet structure is $1 \times 512 \rightarrow 1 \times 16 \times 32 \rightarrow 64 \times 16 \times 32 \rightarrow 128 \times 16 \times 32 \rightarrow 256 \times 16 \times 32 \rightarrow$

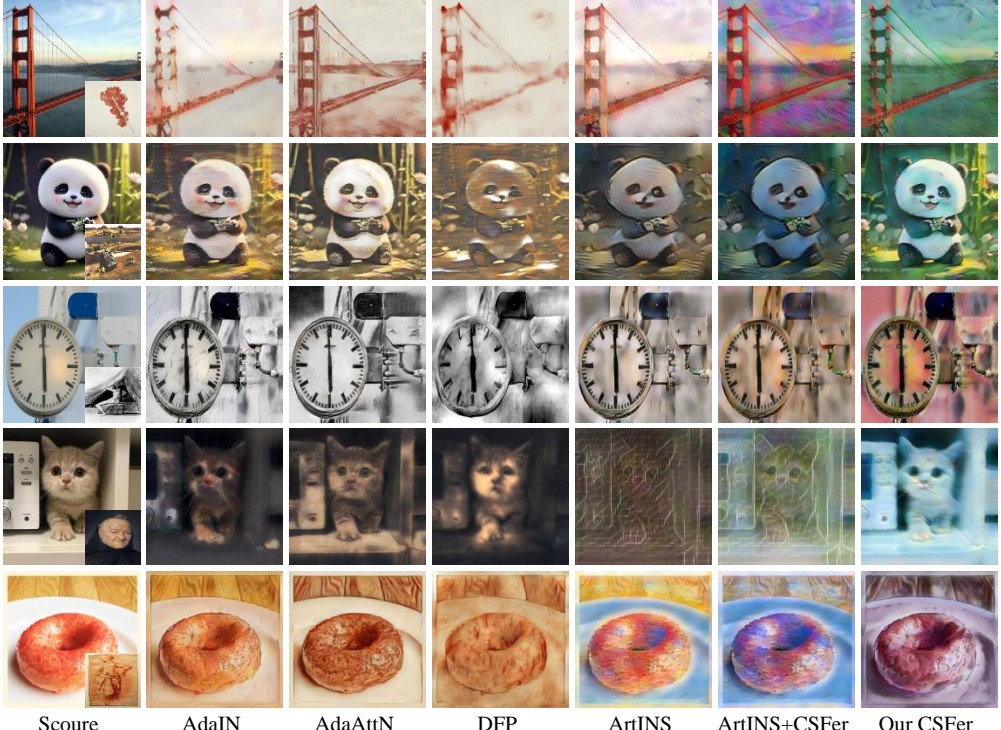

| Scoure | AdaIN | AdaAttN | DFP | ArtINS | ArtINS+CSFer | Our CSFer |

Figure 3: Example style transfer results in comparison with AdaIN (Huang & Belongie, 2017), AdaAttN (Liu et al., 2021a), DFP (Wang et al., 2020) and ArtINS (Xie et al., 2022).

$512 \times 16 \times 32 \rightarrow 512 \times 512$. We conducted our experiments using four NVIDIA GeForce RTX 3090 GPUs. In the style test stage, we consider three datasets: the rest of the WikiArt training set (37,729 style images), the Creative Adversarial Networks (CAN) (Elgammal et al., 2017) (160 style images), and the Creative Decoder (CD) (Das et al., 2020) (select 5 style images from the paper).

**Compared Methods.** To demonstrate that our CSFer can build new and meaningful stylization outcomes, we conduct eleven baselines for CSFer from various aspects. We first consider nine popular artistic style transfer methods, AdaIN (Huang & Belongie, 2017) and LST (Li et al., 2019), AdaAttN (Liu et al., 2021a), ArtFlow (An et al., 2021), StyTr$^2$ (Deng et al., 2022), MST(Zhang et al., 2019) and IEC (Chen et al., 2021a) that transfer arbitrary styles to content images. Recently, deep feature perturbation (DFP) (Wang et al., 2020) tries to produce diverse styles by using orthogonal noise with $\alpha = 0.8$, and ArtINS (Xie et al., 2022) discovers new styles by using independent components of given styles. Furthermore, to verify the generalization ability of our CSFer, we apply the learned PerNet to the LST, MST, IEC and ArtINS methods, called CSFer+LST, CSFer+MST, CSFer+IEC and CSFer+ArtINS. In addition, there are two different methods to create new artistic styles. Creative Adversarial Networks (CAN) (Elgammal et al., 2017) extends generative adversarial networks to generate art. Creative Decoder (CD) (Das et al., 2020) enhances the decoder of variational autoencoder by sampling the neuronal activation patterns.

## 4.2 MAIN RESULTS

We conduct comparisons between our CSFer and both (i) various NST methods and (ii) creative approaches. Subsequently, we demonstrate the generalization capabilities of our CSFer, accompanied by the findings from our user study. Note that all our results exclusively showcase novel stylistic variations while maintaining consistent content. More results, please refer to Appendix D.

**Comparisons with different NST methods.** Various visual examples of style transfer achieved through different methods are illustrated in the Figure 3. We have the following four observations. Firstly, AdaIN (Huang & Belongie, 2017) transfers the artistic style of the style image to the content image with a similar art style, without producing much diversity, not to mention the creativity. Secondly, Although DFP (Wang et al., 2020) generates diverse styles, it still stays close to the given

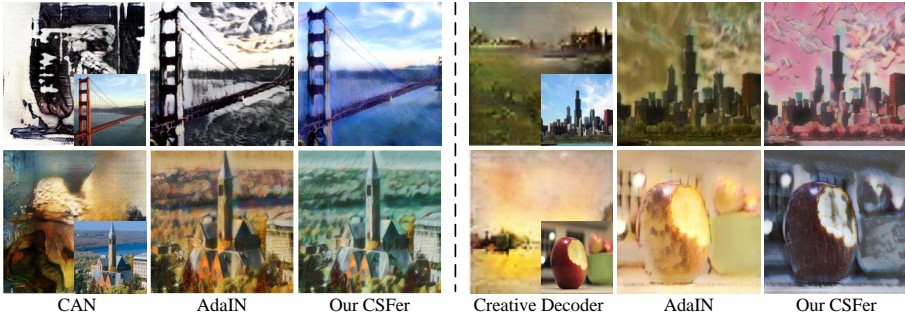

Figure 4: Comparing artistic styles generated by CAN, Creative Decoder, and our CSFer method. AdaIN is used to transfer the styles into the same content images for ensuring a fair comparison.

style due to the small perturbation applied. DFP may fail to transfer the art when using a large perturbation. Thirdly, ArtINS (Xie et al., 2022) produces more diverse styles than AdaIN, LST and DFP, but it has a limit of about 10 bases of the independent components. ArtINS also requires manual parameter setting for the linear combination. Finally, our CSFer, in contrast to these methods, creates novel and meaningful art styles as our neural permutation has the ability to produce novel style bases. Even when applied to ArtINS, ArtINS+CSFer generates richer and more creative results.

*Quantitative aesthetic evaluations.* To ascertain the capacity of our CSFer to generate aesthetically meaningful styles, we employ two methods for aesthetic evaluation, namely, SAAN (Yi et al., 2023) and TANet (He et al., 2022), to gauge the quality of the generated styles. The results are presented in Table 1. It is evident that our CSFer and its various adaptations outperform SOTA style transfer methods in terms of SAAN and TANet aesthetic scores. For instance, ArtINS+CSFer and MST+CSFer achieve the highest SAAN scores, namely 5.112 (WikiArt), 4.746 (CAN), and 4.776 (CD). Similarly, our CSFer and MST+CSFer attain the highest TANet scores, which are 5.444 (WikiArt), 5.521 (CAN), and 5.458 (CD).

**Comparison with the creative methods.** Figure 4 compares the artistic styles generated by CAN (Elgammal et al., 2017), Creative Decoder (CD) (Das et al., 2020), and our CSFer method. To ensure a fair comparison, we used AdaIN to transfer each style to the same content. Our approach produces novel styles with mixed colors, which is a significant improvement over both CAN and the creative decoder. For example, in the second column, CSFer naturally creates a mixed green and red style, which is more visually appealing than the black and yellow styles of CAN+AdaIN. Similarly, in the fifth column, CSFer generates a mixing green and yellow style that is more appealing than the yellow style of CD+AdaIN.

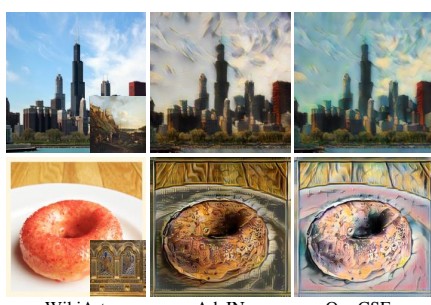

Figure 5: Artistic examples created by our CSFer from the WikiArt dataset.

Furthermore, as shown in Figure 5, our approach can create colorful arts based on the WikiArt dataset, as seen in the results in the last column. In addition, in the first and second columns, our approach create the green and weak pink style from the red and yellow style images, respectively.

**Generalization capability of our CSFer.** Our method exhibits remarkable generalization abilities. It seamlessly integrates with various NST techniques by simply incorporating the acquired PerNet. These methods include LST, MST, IEC, and ArtINS. The visual results, as depicted in Figures 3 and 6, provide compelling evidence that these variations consistently yield fresh and meaningful styles. Additionally, Table 1 presents the aesthetic scores, showcasing how CSFer enhances all the variants except for ArtINS. It's worth noting that comparing ArtINS is challenging due to its linear combination of eight styles, which complicates the tuning of combinatorial parameters. Furthermore, in Figure 7, we present the SPD scores calculated between the original style $I_s$ and both the stylized image $O$ and our creatively stylized image $\widetilde{O}$. This comparison illustrates our CSFer significantly influences the NST methods, resulting in higher SPD scores and novel stylized outcomes.

**User Study.** We conducted a user study to assess the effectiveness of our CSFer algorithm in comparison to other methods. The evaluation was divided into three categories based on different perspectives. These were favorite results among the five NST methods, and creative art results

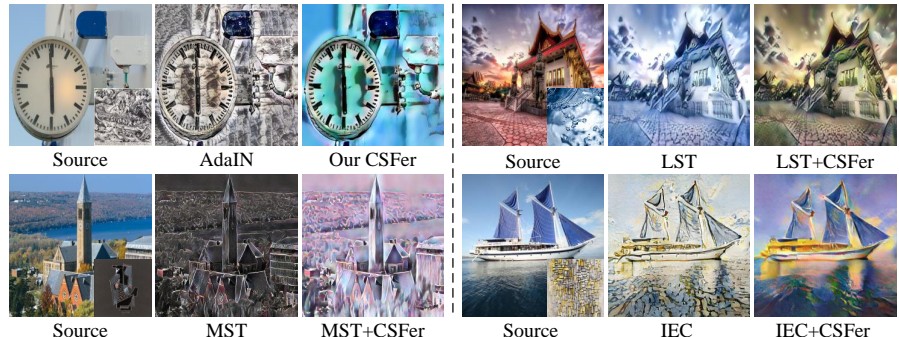

Figure 6: Generalization results using our CSFer to insert into the LST, MST and IEC methods.

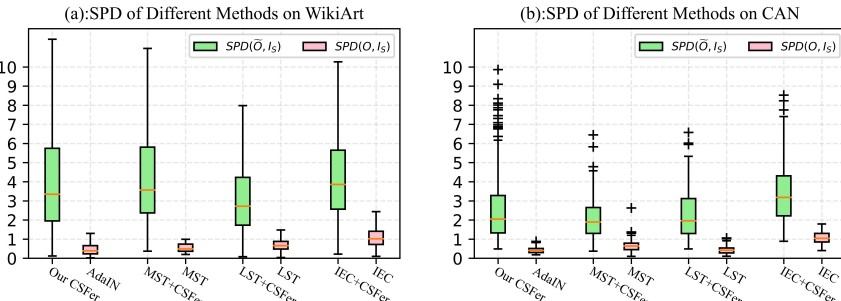

Figure 7: Style Perception Distance (SPD) comparisons with different methods on WikiArt and CAN.

between CAN (or CD) and CS-Fer. Each group received 338 votes, resulting in a total of 1014 votes from 169 users. Ta-

Table 2: Quantitative evaluations of stylization methods.

| Methods | AdaIN | LST | DFP | ArtINS | CSFer | ArtINS+CSFer | CD | CSFer | CAN | CSFer |
|---------|-------|-----|-----|--------|-------|--------------|----|-------|-----|-------|
| Vote ↑ | 64 | 38 | 87 | 16 | **96** | 37 | 146 | **192** | 142 | **196** |

ble 2 reports the voting results. CSFer received the most votes, with $28.4\%$ of users indicating that it was their preferred method, compared to only $4.7\%$ for ArtINS. Additionally, approximately $57\%$ of users believed that CSFer provided a more creative style than the CAN method, and $58\%$ believed that it produced a superior visual experience than CD. Overall, our CSFer was the most popular of all novel creative ways. The scheme of user study, refer to Appendix B

## 4.3 PARAMETER ANALYSIS OF SAMPLING

During our sampling procedure in Figure 2, we determined the values of three parameters $\alpha$, $\beta_1$, and $\beta_2$, in Eq. 6. based on 200 pairs of style and content images. To generate an image set $\widetilde{\mathcal{O}}$ for each image pair, we performed 1000 random permutations of $(\mu_S, \delta_s)$. Our decision to set the parameters $\alpha = 0.2$, $\beta_1 = 0.5$, and $\beta_2 = 0.3$ was informed by Table 3 for $\beta_1$ and $\beta_2$, as well as Appendix C for $\alpha$. These parameter values were selected because they yielded the highest average aesthetic assessment SNNA score (Yi et al., 2023). Using these parameter settings, we obtained a

Table 3: Parameter analysis with $\beta_1$ and $\beta_2$ using the SNNA scores when $\alpha = 0.2$.

| $\beta_1 \backslash \beta_2$ | 0.1 | 0.2 | **0.3** | 0.4 |
|---------|-----|-----|-----|-----|
| 0.3 | 5.054 | 5.157 | 5.214 | 5.244 |
| 0.4 | 5.107 | 5.182 | 5.241 | 5.244 |
| **0.5** | 5.107 | 5.218 | **5.321** | 5.321 |
| 0.6 | 5.150 | 5.233 | 5.233 | 5.319 |

subset of images denoted as $\widetilde{\mathcal{O}}$sub and identified an optimal image $\widetilde{\mathcal{O}}\text{opt}^{(I_s, I_c)}$.

## 5 CONCLUSION

In this paper, we introduce an innovative NST method designed to generate novel and meaningful stylized images using just a single style image. Our approach consists of two crucial components: neural permutation for generating fresh stylized outputs, and the selection of novel and meaningful results from these newly stylized outputs. The former involves training a neural network to rearrange the means and variances of feature maps derived from the style image, while the latter entails evaluating the novelty of these newly stylized outputs to choose the optimal outcome. Our experiments demonstrate that our approach excels in producing novel and meaningful stylized images when compared to both popular NST methods and other creative techniques in the field.

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

Table 4: Parameter analysis with $\alpha$ $\beta_1$ and $\beta_2$ using the SNNA scores.

| $\alpha = 0.1$ | | | | | $\alpha = 0.2$ | | | | | $\alpha = 0.3$ | | | |
|---|---|---|---|---|---|---|---|---|---|---|---|---|---|
| $\beta_1\backslash\beta_2$ | 0.1 | 0.2 | 0.3 | 0.4 | $\beta_1\backslash\beta_2$ | 0.1 | 0.2 | 0.3 | 0.4 | $\beta_1\backslash\beta_2$ | 0.1 | 0.2 | 0.3 | 0.4 |
| 0.3 | 4.961 | 5.068 | 5.121 | 5.154 | 0.3 | 5.054 | 5.157 | 5.214 | 5.244 | 0.3 | 5.095 | 5.187 | 5.245 | 5.286 |
| 0.4 | 5.001 | 5.126 | 5.162 | 5.204 | 0.4 | 5.107 | 5.182 | 5.241 | 5.244 | 0.4 | 5.151 | 5.224 | 5.282 | 5.314 |
| 0.5 | 5.001 | 5.154 | 5.195 | 5.224 | 0.5 | 5.107 | 5.218 | **5.321** | **5.321** | 0.5 | 5.170 | 5.242 | 5.302 | **5.333** |
| 0.6 | 5.091 | 5.179 | 5.224 | 5.255 | 0.6 | 5.150 | 5.233 | 5.233 | 5.319 | 0.6 | 5.183 | 5.271 | 5.318 | **5.352** |

## A  SSIM AND SPD

- **Structural Similarity** (SSIM) (Wang et al., 2004) between images $x$ and $y$ is defined by:

$$\text{SSIM}(x, y) = [l(x,y)]^\alpha [c(x,y)]^\beta [s(x,y)]^\gamma, \qquad (9)$$

where $[l]^\alpha$, $[c]^\beta$ and $[s]^\gamma$ computes the brightness, contrast and structure between images $x$ and $y$, and the parameter is typically set to $\alpha = \beta = \gamma = 1$. SSIM is close to 1 or 0, that is, $x$ is structural similar or dissimilar to $y$.

- **Style Perception Distance** (SPD) (Li et al., 2017a) between images $x$ and images $y$ is computed by:

$$\text{SPD}(x, y) = \sum_{i=1}^{L} \|\mu(\phi_i(x)) - \mu(\phi_i(y))\|_2 + \sum_{i=1}^{L} \|\sigma(\phi_i(x)) - \sigma(\phi_i(y))\|_2, \qquad (10)$$

where each $\phi_i$ denotes a layer in VGG-19 used to compute the style perception distance. In our experiments we use *relu1_1, relu2_1, relu3_1, relu4_1* layers with equal weights. $\mu$ and $\sigma$ represent the mean and variance of each layer. The style of the images $x$ is less similar to the style of the images $y$ as the SPD increases.

## B  THE SCHEME OF USER STUDY

In this subsection, we present the user study, as depicted in Figure 8. The first question involves *Choose your favorite stylized image base on the content image and style image* by considering both the content and style images. This comparison is carried out across various methods, including the work by AdaIN (Huang & Belongie, 2017), AdaAttN (Liu et al., 2021a), DFP (Wang et al., 2020), and ArtINS (Xie et al., 2022). In the second question, we focus on leveraging the CAN dataset (Elgammal et al., 2017) and the Creative Decoder dataset (Das et al., 2020) to generate novel images. Respondents are prompted to select their preferred stylized image based on the content image alone.

## C  PARAMETER ANALYSIS

We determined the values of $\alpha$, $\beta_1$, and $\beta_2$ by examining Table 4. It was observed that when $\alpha = 0.2$, $\beta_1 = 0.5$, and $\beta_2 = 0.3$, our CSFer achieved a relatively high SAAN score, leading us to select these values. Additionally, we visualized the sampling results using these parameters in Figure 9. This visualization demonstrates that our evaluation method is capable of sampling novel and meaningful styles.

## D  MORE VISUAL RESULTS

In this section, we present more visual results. Figure 10 shows more results to verify the generalization abilities of our CSFer method. Figure 11 showcases more visual results compared with the creative methods,*e.g.*, CAN and Creative Decoder.

1.Choose your favorite stylized image base on the content image and style image

Content/Style    ◯ Option A    ◯ Option B    ◯ Option C    ◯ Option D    ◯ Option E    ◯ Option F

Content/Style    ◯ Option A    ◯ Option B    ◯ Option C    ◯ Option D    ◯ Option E    ◯ Option F

2.Choose your favorite stylized image based on the content image

Content    ◯ Option A    ◯ Option B          Content    ◯ Option A    ◯ Option B

Content    ◯ Option A    ◯ Option B          Content    ◯ Option A    ◯ Option B

Figure 8: The process of user study.

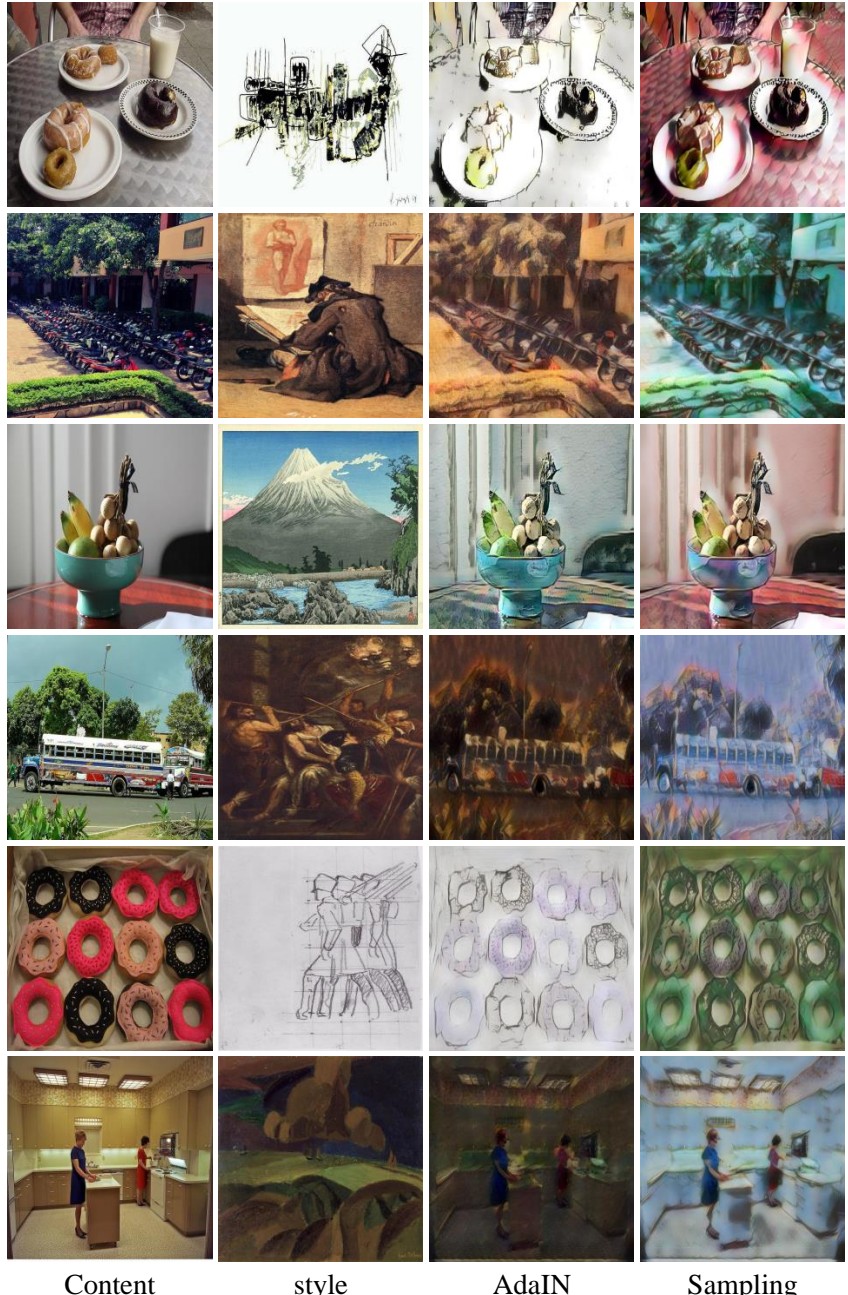

|       Content       |       style       |       AdaIN       |       Sampling       |

Figure 9: Visualizations of the sampling results using the parameters $\alpha = 0.2$, $\beta_1 = 0.5$, and $\beta_2 = 0.3$.

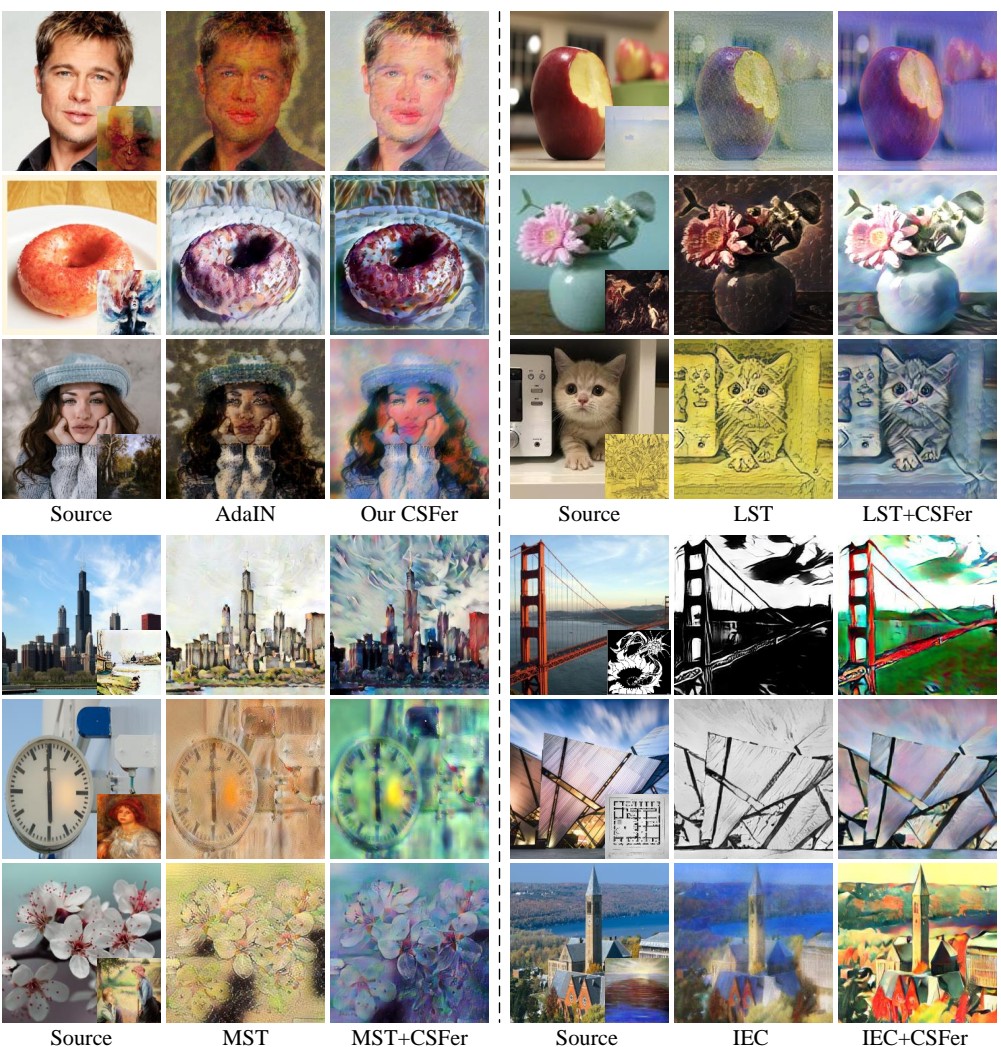

Figure 10: More generalization results using our CSFer to insert into the LST, MST and IEC methods

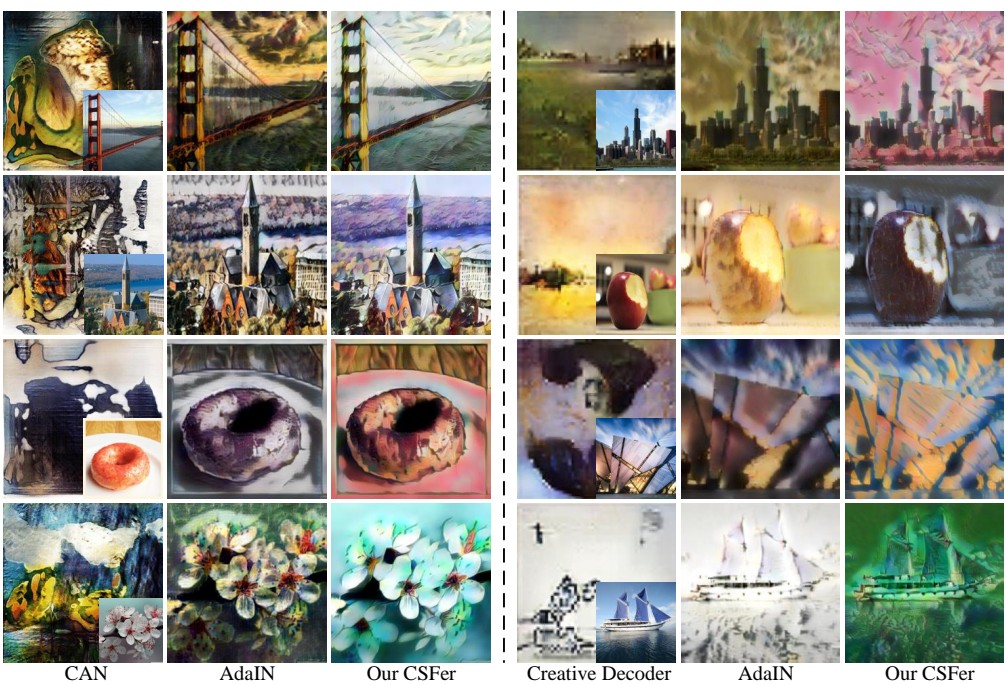

Figure 11: More results by comparing artistic styles generated by CAN, Creative Decoder, and our CSFer method

