# OpenReview forum: "Creative Style Transfer"
_ICLR.cc/2024/Conference — ICLR 2024 Conference Withdrawn Submission_

### Official Review · Reviewer_Wopn · 2023-10-16

**Soundness:** 3 good
**Presentation:** 2 fair
**Contribution:** 1 poor
**Rating:** 1
**Confidence:** 5

**Summary:**

The paper describes a method for style transfer from a content image and a style image. The idea is, rather than to directly mimic to example style, to generate a new style by shuffling feature layers of the style image, and optimize these new styles according to an aesthetic evaluation network.

**Strengths:**

The idea of using an aesthetic evaluation network to judge newly-generated styles is interesting, and, to my knowledge, novel.

**Weaknesses:**

The paper claims that the method creates “exceptional creative artworks” and offers “significant improvement” over previous methods. I disagree. The results look like other neural style transfer methods, with many of the same limitations and artifacts as previous methods. They look unlike real artistic styles in ways that other NST fails to capture real artistic styles, and they do not offer something useful and new.

The paper provides quantitative evaluations that don’t really demonstrate much improvement. The quantitative evaluation in Table 1 is measured in part on the same loss used for optimization, and the scores are very similar to baselines (even with overlapping confidence intervals). A user study is provided, but the description of the evaluation methodology is completely inadequate (no description of participant population, recruitment protocol, questionnaire protocol, randomization, how images were chosen, no statistical analysis of results, etc.), making it impossible to understand what the user study actually measured.

At a more fundamental level, no motivation seems to be provided for the permutation approach. While this is not strictly necessary—it would be a valuable contribution if it worked—it would be better to understand why the authors thought this was an approach worth trying, whether there’s any motivation for this versus any other procedural algorithm one could come up with to make images and/or rearrange networks.

The paper suffers from unscientific and anthropomorphic terminology, using words like “creative,” “meaningful,” “novelty,” “fresh,” and so on. What does the paper mean when they say a style is “meaningful”? What does it mean to say that the goal is to “imaginatively Infuse a fresh and meaningful style”? How is “imagination” and “freshness,” in the everyday senses of these words, embodied algorithmically? Certainly not what I think that most people would understand as “meaningful.” The definitions provided for “creativity” (e.g., Boden et al) are contested definitions that remove the human element from creativity, whereas many psychology studies of creativity focus on human processes. The paper would be better if it used careful terminology rather than anthropomorphic advertising words.

**Questions:**

None

---

### Official Review · Reviewer_sSvh · 2023-10-30

**Soundness:** 1 poor
**Presentation:** 2 fair
**Contribution:** 1 poor
**Rating:** 3
**Confidence:** 5

**Summary:**

This paper introduces a new style transfer problem, dubbed creative style transfer, which aims to imaginatively infuse a fresh and meaningful style from a single style image into the content image. To fulfill the goal, the authors propose to use a neural permutation mechanism to sort the feature mean and variance of the style image by a PerNet. The experiments show that when taking a pair of content and style images as the input, the proposed method can produce a stylized image that does not share the style similarity with the style image while preserving the content of the content image.

**Strengths:**

1. The paper is easy to follow.
2. The results show that the method indeed produces results that are different from the given reference style image.

**Weaknesses:**

1. I do not think the so-called creative style transfer problem is meaningful. When given a pair of content and style images, the task of reference-guided style transfer aims to transfer the style patterns of the style image to the content image, not to create so-called novel style elements from nowhere. It is weird that the proposed method requires the reference style image as the input, while the results have colors and patterns that do not belong to the style image.
2. The neural permutation in the paper does not make sense to me. The original AdaIN method scales and shifts the content image's features by the means and variances of the corresponding style image's features channel-wisely. However, the authors propose to reshuffle the style image's means and variances and then apply them to the content image's features. As the reshuffled means and variances do not come from the same feature extraction branches as those content features channel-wisely, it makes no sense to scale and shift the content features by the reshuffled means and variances. I guess that is why the proposed method can create stylized images that have colors and patterns different from the reference style image.

**Questions:**

1. The PerNet seems to output a single permutation matrix per input style image. So, how can the method produce a set of fresh stylized images by the PerNet?
2. How about pre-calculating and storing the means and variances of a set of style images and then randomly sampling from them to perform AdaIN channel-wisely?
2. Can the authors give some practical applications of the proposed method to validate its practicability?
3. As the proposed method's stylized result does not follow the style image's style patterns (e.g., color and strokes), If the authors want to create novel styles while preserving the content, I think training a generative model to capture the style image distribution, e.g., training a GAN or a diffusion model on the WikiArt dataset, and then randomly sampling from the model while enforcing the output to have the same content as the content image, would be better.

---

### Official Review · Reviewer_D5gN · 2023-10-31

**Soundness:** 2 fair
**Presentation:** 3 good
**Contribution:** 2 fair
**Rating:** 5
**Confidence:** 2

**Summary:**

This paper proposes a new task called creative style transfer, which aims to generate diverse styles from one single style image. The authors propose a neural permutation network to predict the permutation matrix to shuffle the mean and variance vectors of the feature map of the input style image. To supervise the neural permutation network, the authors propose to combine SSIM, Style Perception Distance, and Style-Specific Art Assessment Network as the metric. The proposed method seems to generate a diverse set of styles and achieves some good results both quantitatively and qualitatively.

**Strengths:**

1. This paper proposes a new task called creative style transfer
2. The idea of training a permutation network is interesting
3. The proposed method achieves some good results both qualitatively and quantitatively and also seems easy to incorporate into existing style transfer methods

**Weaknesses:**

**1. Motivation**
- I understand that the goal is to generate diverse (and good) styles. But if we use the style perception distance to select outputs that are not that similar to the provided style image, should we still call it “style transfer”? It seems to me that we can use the same metric to train a model to permeate the mean and variance of the content image and still give you some good results. Or we can even use the mean and variance as input to the model to predict some mean and variance, without even needing the style image?
- Thus, I feel that the main goal of this paper is slightly different from previous style transfer methods. And I am not sure if the metrics in Table 1 are the proper ones.

**2. Others**
- How to decide the values of K1 and K2?
- What if we don’t use position encoding?
- Will the results be very different if use a greedy algorithm instead of the Hungarian algorithm when converting the soft permutation matrix into a hard permutation matrix?
- The user study seems a bit problematic. For the first question, the style image is too small which seems to encourage the users to ignore them and focus on whether the output is visually better (and thus no need to use a style image as a reference). For the second question, whether the output image looks good highly depends on the input style image. Using multiple style images as input and selecting the one with the highest SSAN score as the baseline seems more reasonable and fair to me.

**Questions:**

Please see the weakness section.

---

### Official Review · Reviewer_rMps · 2023-11-02

**Soundness:** 1 poor
**Presentation:** 1 poor
**Contribution:** 1 poor
**Rating:** 1
**Confidence:** 5

**Summary:**

The authors permute distrubtions of vectors responses, claiming that doing so leads to creative style transfer.

**Strengths:**

I like the aim of the paper.

**Weaknesses:**

The method of permuting distributions can nver and will never be creative: the method is nut suitable for the aim.

That paucity of the model  can be seen from the output - which looks almost exactly the same as the majority of NST that has gone before it (and most of which models the artistic process very badly).

The experimental section uses automatic measures that fail to capture creativity,
and the experiment that involved people merley asked them about a preference.
Which means the experiment is uniformative with respect to the hypothesis (that permuting permutations is creative).

**Questions:**

What is the principle of creativity you use to ustify your model of permuting distributions?

Why did your experiment with people ask about preference rather than creativity?

Is you loss function a good model of the way people create art?

**Details Of Ethics Concerns:**

no concern